# Urbanization Paradox of Environmental Policies in Korean Local Governments

**Yongrok Choi [1], Hyoungsuk Lee [2,*], Hojin Jeong [3,*] and Jahira Debbarma [3]**

1    Department of International Trade, Inha University, Incheon 22212, Republic of Korea
2    Energy Environment Policy and Technology, Graduate School of Energy and Environment
     (KU-KIST Green School), Korea University, Seoul 02841, Republic of Korea
3    Program of Industrial Security Governance, Inha University, Incheon 22212, Republic of Korea
*    Correspondence: lhs2303@korea.ac.kr (H.L.); jdilla@inha.edu (H.J.)

**Abstract:** Many developing countries have been experiencing the problems of urbanization, particularly regarding carbon emission and polluted air emission mitigation. Is it possible to simultaneously achieve these two different clean and green economic strategies? This study analyzes this paradoxical issue of air pollution in terms of $PM_{2.5}$ efficiency. To evaluate the performance of regulatory policies on air pollution and to find out the governance factors, this paper adopts the stepwise approach. In the first stage, we evaluate the cross-sectional $PM_{2.5}$ efficiency of 16 Korean municipalities for the period between 2012 and 2017 and determine whether this performance is sustainable using the Malmquist Productivity Index (MPI). We concluded that most local governments lack sustainable governance on regulation policies for clean air. Using the Tobit model in the second stage, this study showed that regional economic development (GRDP) and an patent for clean air technology innovation are the most important strategic factors that promote sustainability in regulation policy performance.

**Keywords:** $PM_{2.5}$ efficiency; policy paradox; local government; governance

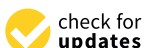



## 1. Introduction

Until the mid-2010s, Korea had never had significant air pollution issues, so the focus of the environmental policies has always been on the abatement of greenhouse gases (GHG) [1]. As shown in Figure 1, until 2014, air pollution was low and stable, resulting in no emphasis on clean air. Based on this policy paradigm, there have been strong promotional policies for diesel cars in Korea, such as tax incentives on diesel and subsidies for the purchase of diesel trucks, which substitute oil cars with diesel ones to reduce $CO_2$ emissions. As diesel cars produce far fewer carbon dioxide emissions than oil-fueled ones, the government is assured that diesel cars are the best alternative to meet the challenges of global warming and the climate crisis. As a result, diesel cars are rapidly becoming the major transportation device and the optimal solution for small-businesses, which use diesel trucks, such as mini-cargo delivery and/or street food trucks. Moreover, the government exempted the annual environmental improvement charge of 240,000 won for diesel cars. Ten years later, diesel cars have led to significant environmental problems, such as heavy air pollution, instead of $CO_2$ emissions. Diesel cars emit very little carbon dioxide, but they emit a lot of nitrogen dioxide instead. The $PM_{2.5}$ created from this $NO_2$ led to uncontrollable air pollution, particularly during the winter season, in a society with a strong sustainability focus. $PM_{2.5}$ can easily enter the blood vessels through the alveoli, causing inflammation and resulting in cardiovascular diseases, such as angina and stroke. Even more dangerously, $PM_{2.5}$ can accumulate in the lungs and blood over time, resulting in serious illnesses ([2], p. 3). Soon, the Korean government began to realize that air pollution may become a serious social problem and a different environmental challenge from the climate crisis. Many Koreans began to realize that $PM_{2.5}$ is a serious environmental issue

that has a strong impact on the quality of life. $PM_{2.5}$ became a major concern, and diverse papers on the topic published in China received worldwide attention, including in Korea.

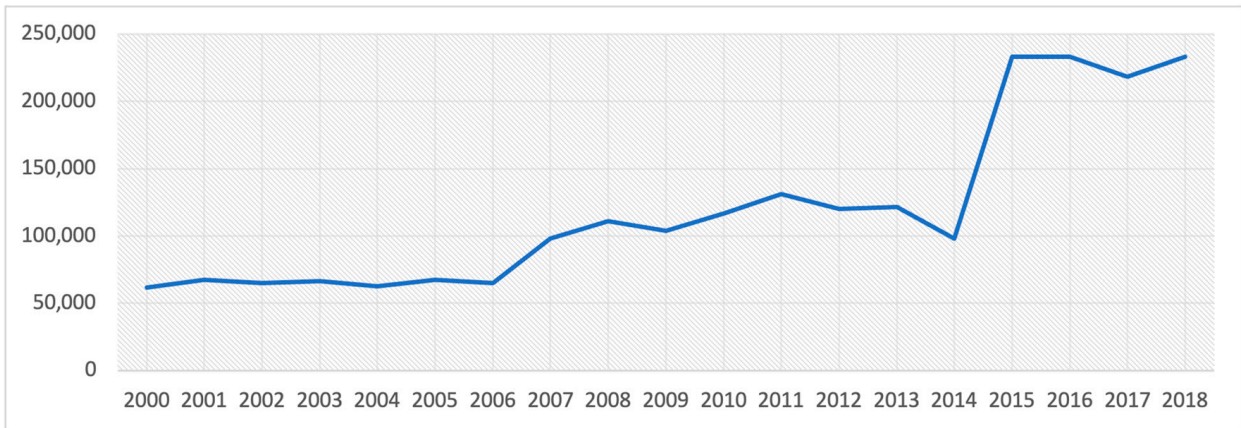

**Figure 1.** Air pollution (PM10) trend (Ton) [1].

In Figure 1, the air pollution trend of particulate matter less than 10.0 μm (PM 10) clearly shows that the level of air pollution remained low until 2006, before rapidly increasing and capturing public attention. Both the central and local governments began to take measures to curb this trend. Nonetheless, the policy effects of local governments were not consistent with each other, resulting in high levels of air pollution. Compared with the level of air pollution recommended by the World Health Organization (WHO), the current level is around three to four times higher, affecting more than 23,000 people annually [3]. KBS News, Korea's public broadcasting system, emphasized that PM levels had more than a tenfold effect on the expected life span of Korean people.

Figure 2 represents the research of Jeonnam, Chungnam, Ulsan, and Gyeongbuk, which indicates high air pollution levels measured by $PM_{2.5}$ in provinces with industrial complexes and power plants. Jeju Island showed stable and low levels of air pollution because of its policies to promote its clean and unpolluted natural heritage. Nevertheless, the national level of air pollution did not improve because of a lack of governance on several policies. For example, the Gyeonggi province barely managed to decrease its level of polluted air, despite a strong emphasis on regulations.

Another issue is that managing air pollution requires different measures than managing the climate crisis, and there are paradoxical difficulties to aligning the different policy directions that are required to achieve both in order to create an eco-friendly economy, as shown by the diesel car case. Each local government has made numerous efforts to solve these problems in a harmonized way. However, Figures 1 and 2 show that most of these efforts have been unsuccessful, not due to the lack of regulation policies, but due to the lack of governance on the sustainable implementation of the policies. Why has the emission of carbon dioxide and $PM_{2.5}$ increased in most local governments? Even if some local governments, such as Jeonnam, can demonstrate short-term success in managing $PM_{2.5}$, is it possible for these policies to be sustainable? If not, what should the local government do to harmonize $PM_{2.5}$ and carbon dioxide management? This creates a policy paradox, as local governments must simultaneously focus on solving two vastly different problems. With this in mind, this study aims to suggest policies that will align the short-term effect of $PM_{2.5}$ policies and the long-term or sustainable effect of $CO_2$ regulation. Therefore, the purpose of this research is to help local governments manage the air pollution curve in Korea.

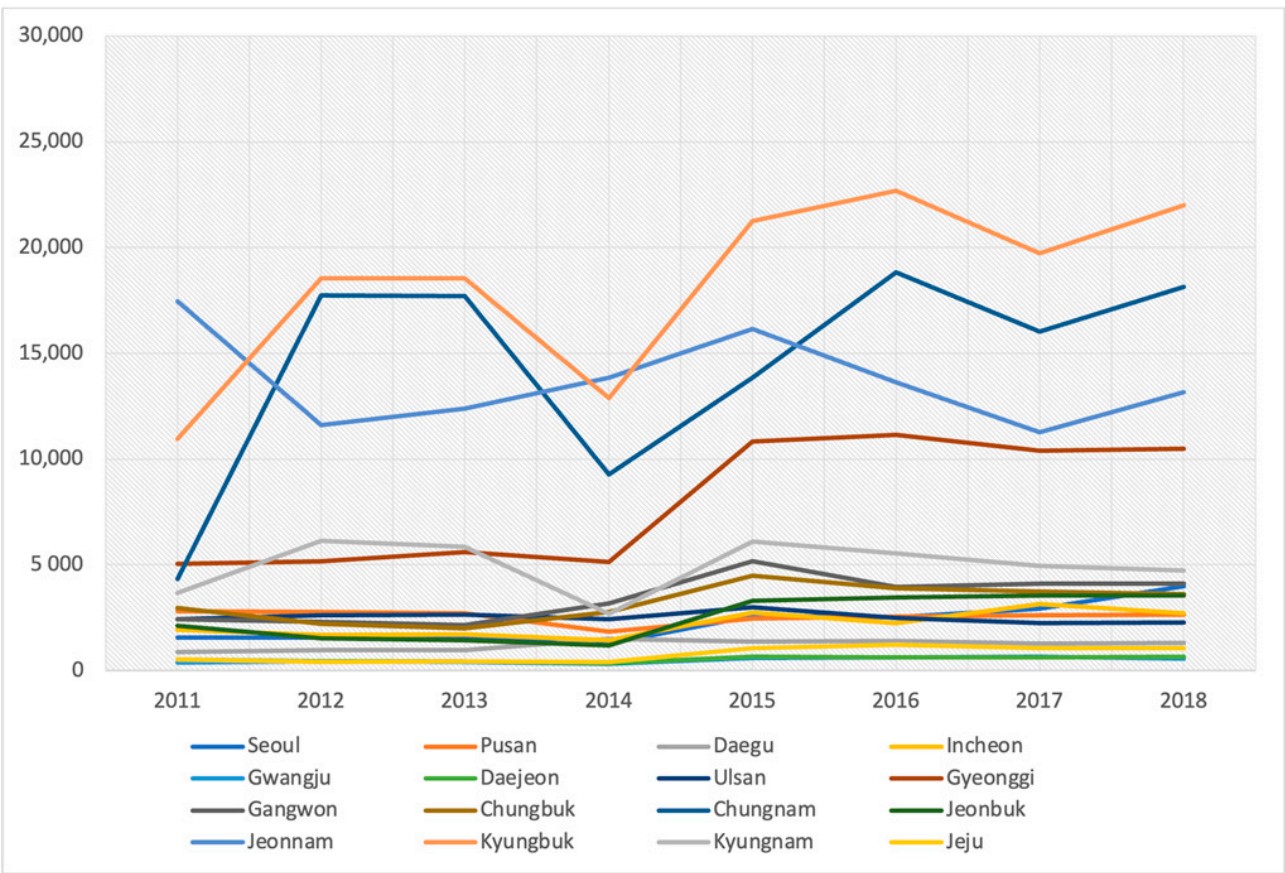

**Figure 2.** Air pollution trend (PM$_{2.5}$) in local governments (Ton) [1].

To examine these research questions, the first stage of this study evaluates the PM$_{2.5}$ efficiency of 16 local Korean provinces to show the benchmarking candidate for the regulation and promotion policies of more competitive local governments. However, efficient policies of successful local government do not necessarily result in sustainable performance, so the second stage of our research analyzes the dynamic productivity of PM$_{2.5}$ for the 16 provinces. Based on these cross-sectional and dynamic evaluations, the third part of the study strives to find the successful governance factors using Tobit analysis. For this purpose, Chapter 2 will present our empirical model, which is based on a literature review.

## 2. Literature Review

### 2.1. Models and Variables of the Environemntal Efficiecny

For the first stage, multi-inputs and outputs shall be employed to evaluate the coupling or decoupling effects on energy and environment (E&E) analysis, based on the comparative results in Table 1. [4–13] Since the data envelope analysis (DEA) is the leading choice for handling multi-inputs/outputs models, our research utilizes this approach. It does not need any specific form of the production function for this multi-inputs/outputs model. Thus, it is advisable to evaluate the relative efficiency for all decision-making units (DMUs).

**Table 1.** Literatures Comparison for the model and variables.

| Author(s) | Research Subject | Measurement | Methodology | Variables |
|---|---|---|---|---|
| Lee et al., (2018) [4] | GHG performance in Korea. | Energy efficiency | DDF | Input—energy, capital, labor Desirable output—GRDP Undesirable output—Greenhouse gas |
| Na wang et al., (2019) [5] | Compare the city-level environmental efficiency. | Environmental efficiency | GMNDDF | Input—capital, labor, energy Desirable output—GRDP Undesirable output—$CO_2$ |
| Chang et al., (2013) [6] | Analyze the environmental efficiency of China's transportation. | Environmental efficiency | SBM-DEA | Non-energy input—labor, capital Energy input—energy Desirable output—value-add Undesirable output—$CO_2$ |
| Choi et al., (2020) [7] | Atmospheric environmental efficiency in China. | Atmospheric efficiency | SBM-DEA | Input—labor, capital, energy. Desirable output—GRDP Undesirable output—SO2, NOx, PMs |
| Ning Zhang et al., (2013) [8] | Environmental energy efficiency of China. | Energy efficiency | SBM-DEA | Input—capital, labor, energy Desirable output—GDP Undesirable output—$SO_2$, $CO_2$, COD |
| Yang and Lee (2022) [9] | $CO_2$ emission efficiency in China. | $CO_2$ emission efficiency | pZSG-DEA | Input—population, capital, energy Desirable output—GRDP Undesirable output— $CO_2$ |
| Chen et al., (2015) [10] | Environmental efficiency in China. | Environmental efficiency | DEA | Input —energy, social fixed assets investment Desirable output—GDP Undesirable output—wastewater, solid, gas |
| Aviles-Sacoto et al., (2021) [11] | Environmental performance evaluation in Mexico. | Environmental performance | DEA | Input—green investment, renewable energy Desirable output—total water, $PM_{2.5}$ |
| Wang et al., (2021) [12] | Total factor energy efficiency in China. | Energy efficiency | DEA | Energy input—electricity, natural gas, artificial gas, industrial fuel oil Economic input—capital, total urban employment Output—GDP, retail sales of consumer goods, budgetary revenue of local government |
| Yu et al., (2022) [13] | $PM_{2.5}$ performance in China. | Environmental performance | DEA | Input—capital, labor, energy. Desirable output—GDP Undesirable output—$PM_{2.5}$ |

From this perspective, Lee et al. [4] examined the greenhouse gas efficiency in Korea using the Directional Distance Function (DDF) model with the input variables, capital, labor, and energy consumption, together with GRDP and greenhouse gases, respectively. With respect to China, Ning Zhang et al. [8] examined the environmental energy efficiency of China, choosing capital, labor, and energy as the input variables, along with GDP and $SO_2$ as the desirable, and $CO_2$ and COD as the undesirable, output variables. Choi et al. [7] examined the atmospheric environmental performance of 30 major cities in China using the SBM-DEA with a meta-frontier model. Three pollution matters (SO2, NOx, PMs) were included in this study.

These studies commonly discovered that Seoul and Ulsan in Korea, and Guangdong and Shanghai in China, are efficient DMUs. This stems from the fact that these four cities have easy access to capital and technology, as more advanced regions may result in more environmental-friendly economic development. Our research question, therefore, is as follows: Can DMUs with a developed economy be significantly more efficient in terms of $PM_{2.5}$ efficiency?

Diverse methodological approaches, based on DEA, have tried to answer this question. Yang and Lee [9] used population, capital, and energy as input variables, with GRDP and $CO_2$ as undesirable output variables, by employing a prospective zero-sum gain data envelopment analysis (pZSG-DEA) to examine the $CO_2$ emission efficiency in China. Diverse inclusion, however, did not bring precise implications and suggestions on the specific effect on DMUs. As the original DEA model takes a radial approach, which may lead to overestimating and comparatively low discriminating power by ignoring slack variables, the SBM-DEA has been used as an alternative to traditional DEA to seizure the entire feature of inefficiency regarding variables' slacks in the efficiency measures [6–9]. Moreover, the SBM-DEA approach gives more reliable and appropriate implications and suggestions because of its simple but precise setting of the production technology [7–9]. Therefore, we apply the SBM-DEA alongside the undesirable output of air pollution to determine the environmental efficiency evaluation. Based on the comparative result shown in Table 1, we selected three input variables of labor, capital, and energy consumption, and two outputs of GRDP and $PM_{2.5}$ emissions, to measure efficiency.

As the objective of the research is to find the best governance factors for specific local governments in Korea that contribute to managing air pollution, namely $PM_{2.5}$, our methodology is based on a three stepwise approach. In the first stage, we measure the $PM_{2.5}$ efficiency of 16 local Korean governments by using the SBM-DEA. However, the model cannot measure efficiency changes over time and thus cannot determine the sustainability of the performance. Here, governance shall be defined as the workable mechanism for the sustainable performance. Thus, to measure the total factor productivity (TFP) growth between different periods, the Malmquist productivity index (MI) will be adapted, resulting in the governance levels of the local governments. As our objective is to find the factors or strategic determinants to better manage air pollution, we need explore the determinants of the efficiency value by using the Tobit analysis to enhance the $PM_{2.5}$ efficiency of these provinces in the second stage.

### 2.2. Determninats of the Environemntal Efficiecny: Tobit Approach

For the second stage, we need to find out the determinants of the efficiency. To evaluate these non-negative values of efficiency, there is diverse research on the Tobit model regarding the role of the determinants in effecting the limited values of the outputs. As shown in Table 2, many studies in the E&E field have adopted the Tobit model to find out the appropriate determinants on the environmental efficiency [14,15]. For example, Deng et al. [16] concluded that government expenditure is important for promoting environmental-friendly energy efficiency, while Debbarma et al. [17] also confirmed that credit access by the government would promote environmental efficiency, implying the key role of the government in the better governance on the environmental efficiency.

Based on the argument on the importance of the governance of policies, Table 2 shows the common factors related with the government policies in detail. First of all, many studies [14–20] have adopted GRDP as an economic factor. As economic level is highly related to its environmental performance, we also adopted GRDP. Secondly, regional characteristics, such as urbanization rate or population density [14,15,17,18], are also widely adopted to explore environmental efficiency. We selected population density and greenbelt width, instead of urbanization rate. Thirdly, because the level of $PM_{2.5}$ emission is tied to the energy structure, we selected renewable energy for it. Lastly, patent is added to find out the regional innovation effect on environmental efficiency.

**Table 2.** Research on the application of Tobit model.

| Reference(s) | Field of Research | Variables |
|---|---|---|
| Chen et al., (2017) [14] | Environmental energy efficiency in the Yangtze River Economic Zone in China | GDP per capita<br>Environmental investment<br>Population density<br>Foreign trade degree<br>Industrial structure |
| Gong et al., (2022) [15] | Health resource allocation efficiency in Sichuan | GDP per capita<br>The average annual income of residents<br>Urbanization rate<br>Population density<br>Education |
| Deng et al., (2020) [16] | Efficiency in the logistics industry in China | Economic development<br>Logistics efficiency<br>Energy structure<br>Government expenditure |
| Debbarma et al., (2021) [17] | Efficiency in agriculture industry in India | Land<br>Livestock<br>Fertilizers<br>Agricultural cultivation<br>Urbanization rate<br>Average rainfall<br>Export<br>Credit access |
| Ma et al. [18] | Ecological efficiency in China | GRDP per GDP<br>Degree of openness<br>Intensity of R&D expenditure<br>Urbanization rate<br>The proportion of energy consumption<br>The proportion of investment in environmental pollution control |
| Zhu et al. [19] | Eco efficiency of China in industrial investment | GDP<br>Investment in treatment of industrial pollution<br>Foreign direct investment<br>Research and development expenditure<br>Total education funds<br>Total import and export trade |
| Aldieri et al. [20] | Energy efficiency of 148 developing and transition countries | Renewable energy<br>R&D<br>Outages<br>Generator<br>Days Connection |

These papers argued that the lack of governance resulted in lower efficiency. Therefore, we should focus on the strategic factors to promote strong governance by applying the Tobit model.

As shown in Table 1, previous research has extensively investigated environmental efficiency based on $CO_2$ emission. However, very few studies have considered $PM_{2.5}$ as an undesirable output. Moreover, most of the research has focused on the efficiency level of DMUs, not on their governance factors. Therefore, our attempt to measure the $PM_{2.5}$ efficiency of Korean local governments based on the region-level data from 2012–2017, will shed light on the optimal path control of clean and green growth in developing countries.

### 3. Methodology and Data

*3.1. PM$_{2.5}$ Efficiecny*

To examine the PM$_{2.5}$ efficiency, the commonly used DEA concept will be utilized with the undesirable output of PM$_{2.5}$ [21]. The DEA model is divided into two forms: radial and non-radial. The radial approach's inputs and outputs may lack information regarding the inactive (or neglected) efficiency of the variables involved in the production process adjusted to the efficiency goal in the same proportion [22]. In order to avoid this proportionate bias, the non-radial approach employs the slack variable, resulting in a high discriminating power and an unbiased estimate. As we aim to assess the PM$_{2.5}$ performance of local governments in Korea precisely, we adopt the undesirable SBM-DEA [6–9].

Assume *n* decision-making units (DMUs), with each DMU consumed *m* inputs, which generate $g_1$ (desirable output) and $b_2$ undesirable (output). The three factors' vectors are defined as $X \in R^m$, $Y^g \in R^{g_1}$ and $Y^b \in R^{b_2}$, and the matrices X, $Y^g$ and $Y^b$ could be expressed as follows [23]:

$$X = [x_1, x_2, \ldots, x_n] \in R^{m \times n},$$
$$Y^g = y_1^g, y_2^g, \ldots, y_n^g, \in R^{g_1 \times n}, \text{ and}$$
$$Y^b = y_1^b, y_2^b, \ldots, y_n^b, \in R^{b_2 \times n}$$
$$X, Y^g \text{ and } Y^b > 0$$

Here, *m* implies the number of inputs consumed by $i^{th}$ local government (**DMU$_i$**) during the production process; $g_i$ is the number of desirable outputs and $b_i$. The production possibility set (*P*) under CRS (constant return to scale) condition is described in Equation (1) [24]:

$$P = \left\{ \left( X, Y^g, Y^b \right) \right\} | x \geq X\lambda, y^g \leq Y^g\lambda, y^b \geq Y^b\lambda, \lambda \geq 0 \} \tag{1}$$

The SBM-DEA model with both the desirable and undesirable outputs is described as Equation (2) [25]:

$$p^* = \min \frac{1 - \frac{1}{m}\sum_{i=1}^m \frac{s_i^-}{x_{i0}}}{1 + \frac{1}{g_1 + b_2}\left( \sum_{r=1}^{g_1} \frac{s_r^g}{y_{r0}^g} + \sum_{r=1}^{b_2} \frac{s_r^b}{y_{r0}^b} \right)} \tag{2}$$

$$\text{s.t.} \begin{cases} x_0 = X\lambda + s^-; y_0^g = Y^g\lambda - s^g; y_0^b = Y^b\lambda + s^b \\ s^- \geq 0, \ s^g \geq 0, s^b \geq 0, \ \lambda \geq 0 \end{cases}$$

where $\lambda$ denotes the weight vector, $s^-$ the slacks of inputs, and $s^g$ and $s^b$ the slacks of the desirable and undesirable outputs in Equation (2). The vector $s^-$ illustrates the overused inputs. $s^b$ indicates the overproduced undesirable outputs (PM$_{2.5}$ emission in this study), and the vector $s^g$ represents the shortage of desirable outputs (GRDP in this study). Further, the subscript denotes the estimated DMU in the model. If $\rho^* = 1$, a local government is regarded to be efficient with all the slack variables equal to zero ($s^{-*} = s^{g*} = s^{b*} = 0$), even if there are undesirable outputs; and vice-versa for $\rho^* < 1$. For a local government to become efficient, it must eliminate excess inputs and undesirable outputs while increasing and adjusting the deficit of desirable outputs.

*3.2. Tobit Regression Model*

The expected value of the residuals is necessarily zero in the OLS hypothesis, resulting in inconsistent or biased estimates for the censored or truncated data [26,27]. As the efficiency value in this study is measured between 0 and 1, the OLS is an inappropriate way to evaluate the efficiency value. Owing to this concern, the Tobit regression model is regarded as a suitable model to solve problems with limited dependent variables [28]. Tobin first proposed the Tobit regression model in 1958 to alleviate zero-inflated data for

observations of household spending on durable goods [21]. This model has since been widely used in environmental studies [29]. The model is defined as Equation (3) [30]:

$$Y_{np} = \begin{cases} Y_{np} = \beta^T x_{np} + \epsilon_{np} \ \ \beta^T x_{np} + \epsilon_{np} \\ \\ 0 \ otherwise \end{cases} \tag{3}$$

where the output variable is $Y_{np}$, with the explanatory variable $x_{np}$; $\beta^T$ is the vector of the regression coefficient of the explanatory variable; and $\epsilon_{np}$ is the stochastic error with the distribution of N(0, $\sigma^2$). To evaluate the determinants of efficiency in local governments, the Tobit model is expressed as follow [27]:

$$Y_{np} = \beta_0 + \beta_1 Z^1{}_{np} + \beta_2 Z^2{}_{np} + \beta_3 Z^3{}_{np} \ldots \beta_x Z^X{}_{np} + \epsilon_{np} \tag{4}$$

where $Y$ is the PM$_{2.5}$ efficiency of local government; $np$ presents the $n^{th}$ local government and the period of study; $\beta_x$ is the coefficient; $Z^X{}_{np}$ is the explanatory variable; $\epsilon_{np}$ denotes the stochastic error.

*3.3. Data*

As our DMUs are provincial authorities, our inputs of capital, labor, and energy consumption as the quantities utilized in each province are shown in Table 3. The desirable output is the local GDP (GRDP), and the polluted air of PM$_{2.5}$ produced in each province is the undesirable output. Their statistical characters are shown in Table 3, and the locations of the provinces are shown in Figure 3. Some provinces, such as Seoul, Busan, and Ulsan, are well-known for their dynamic economic performances. Others, such as Gangwon and Jeju, are famous for their environmental protective atmosphere. The majority of the previous papers showed the coupling effect led by the economic performance with environmental protection represented by carbon emission and/or greenhouse gases, implying that Seoul is a leading city for environmental efficiency, while Gangwon lags behind. However, in the case of air pollution measured by PM$_{2.5}$, the results are not conclusive. Thus, to determine the performance of the clean air policies in these provinces, we performed the first stage of empirical tests.

**Table 3.** Descriptive statistics of input and output variables, 2012–2017.

| Variables | Unit | Mean | St Dev | Minimum | Maximum |
|---|---|---|---|---|---|
| Labor | $10^3$ persons | 1621.05 | 1621.58 | 303.00 | 6685.00 |
| Capital | $10^9$ Won | 28,108.81 | 26,681.18 | 3827.10 | 147,245.50 |
| Energy consumption | $10^3$ t oil-eq | 13,419.36 | 11,540.89 | 1087.00 | 41,611.00 |
| GRDP | $10^9$ Won | 101,216.03 | 105,692.09 | 13,193.14 | 451,426.42 |
| PM$_{2.5}$ | $10^5$ t PM$_{2.5}$ | 14,519.84 | 11,281.71 | 1346.36 | 41,229.49 |

Source: Korea Official Statistics Information System (KOSIS) (https://kosis.kr), accessed on 6 October 2022.

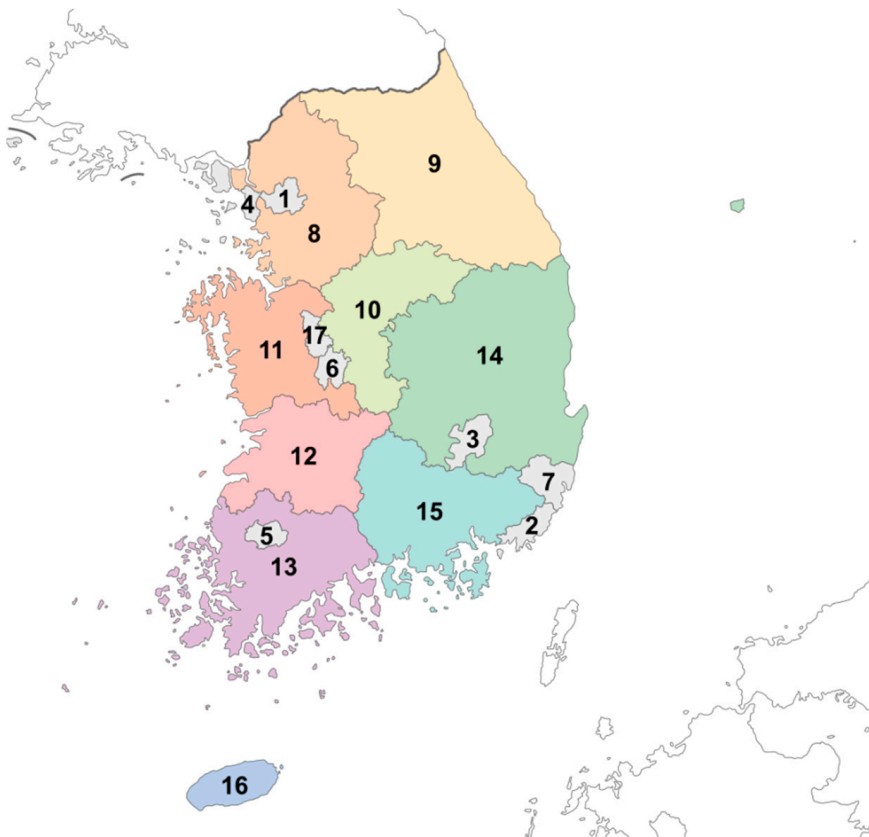

**Figure 3.** Map of 17 local governments in South Korea.

## 4. Result and Discussion

### 4.1. Empirical Results of PM$_{2.5}$ Efficiency and Malmquist Productivity in Local Government

Based on the Equation (2), we obtained the PM$_{2.5}$ efficiency of 16 Korean local governments. The value is from 0.3235 to 1 for 6 years, with an average score of approximately 53.7% (see Table 4). The average score indicates that there is a potential efficiency improvement of 46.3% in Korea at the optimal frontier. Due to the lack of data, the 17th municipality in Figure 3, Sejong, the administrative capital of Korea, has not been included in our research. With respect to the nationwide average of PM$_{2.5}$ efficiency, it shows an increasing trend over time, particularly after 2015. (see Figure 4). Nonetheless, the performance of PM$_{2.5}$ efficiency differs widely, as shown in Table 4.

It is noteworthy that with a similar economic performance regarding their GRDP, Seoul and Busan show very different performances in terms of their PM$_{2.5}$ efficiency. Why does this happen? Is it possible for Seoul to take a leading role over time? To answer these questions, we must evaluate the PM$_{2.5}$ productivity over time. As we are examining the sustainability of air pollution policies in local governments from a dynamic perspective, we should analyze the technological change in productivity instead of simply the efficiency. According to Choi et al. [7], to evaluate the dynamic change in the production technology frontier over time, we must first define the Malmquist Productivity Index (MPI) as the ratio between two periods. As shown in Figure 5, Seoul is not a leader in the dynamic perspective of productivity. The productivity by MPI dropped rapidly in Seoul, particularly in 2015. In contrast, Chungnam showed a strong peak in productivity growth in 2017, despite its low PM$_{2.5}$ efficiency. To solve this coupling issue between clean air and economic development, the government should maintain a sustainable performance in air pollution policies. To find out the governance factors in such policies, the Tobit model is utilized with potential variables, based on Table 2.

**Table 4.** PM$_{2.5}$ efficiency of 16 local governments, 2012–2017.

| No. | Local Government | 2012 | 2013 | 2014 | 2015 | 2016 | 2017 |
|---|---|---|---|---|---|---|---|
| 1 | Seoul | 1 | 0.9423 | 1 | 0.9422 | 0.9831 | 1 |
| 2 | Pusan | 0.4662 | 0.4563 | 0.5344 | 0.5334 | 0.5262 | 0.5405 |
| 3 | Daegu | 0.5279 | 0.4998 | 0.4946 | 0.5106 | 0.5155 | 0.5456 |
| 4 | Incheon | 0.3735 | 0.3722 | 0.4161 | 0.3905 | 0.4209 | 0.4091 |
| 5 | Gwangju | 0.5959 | 0.5958 | 0.6631 | 0.6595 | 0.6601 | 0.6635 |
| 6 | Daejeon | 0.6091 | 0.5983 | 0.6486 | 0.6273 | 0.6457 | 0.6498 |
| 7 | Ulsan | 1 | 0.9366 | 0.9213 | 1 | 1 | 1 |
| 8 | Gyeonggi | 0.5641 | 0.5848 | 0.6177 | 0.5844 | 0.5928 | 0.6393 |
| 9 | Gangwon | 0.2899 | 0.3158 | 0.3237 | 0.34 | 0.3682 | 0.3801 |
| 10 | Chungbuk | 0.3986 | 0.4147 | 0.4171 | 0.4375 | 0.4623 | 0.4883 |
| 11 | Chungnam | 0.4961 | 0.443 | 0.4265 | 0.4766 | 0.5082 | 0.6935 |
| 12 | Jeonbuk | 0.3687 | 0.3861 | 0.4113 | 0.3769 | 0.3716 | 0.3913 |
| 13 | Jeonnam | 0.3235 | 0.335 | 0.3264 | 0.3382 | 0.3538 | 0.3662 |
| 14 | Kyungbuk | 0.3673 | 0.3816 | 0.401 | 0.396 | 0.4058 | 0.4159 |
| 15 | Kyungnam | 0.4608 | 0.4624 | 0.503 | 0.4897 | 0.4961 | 0.4942 |
| 16 | Jeju | 0.4719 | 0.4452 | 0.487 | 0.4756 | 0.4849 | 0.4967 |
|  | Average | 0.5196 | 0.5106 | 0.5370 | 0.5362 | 0.5497 | 0.5734 |

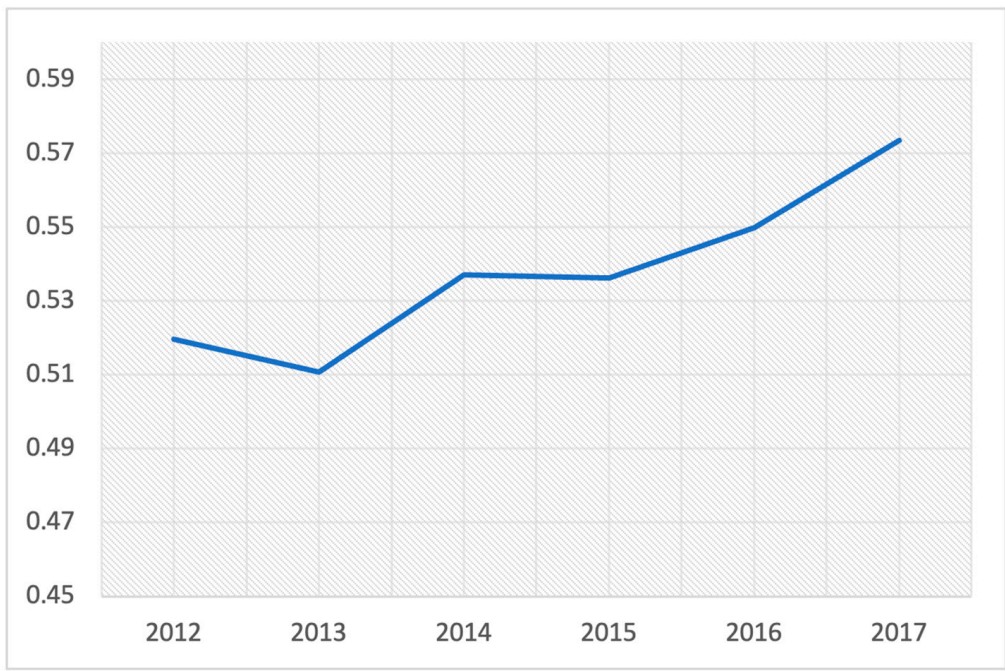

**Figure 4.** National PM$_{2.5}$ efficiency's trend, 2012–2017.

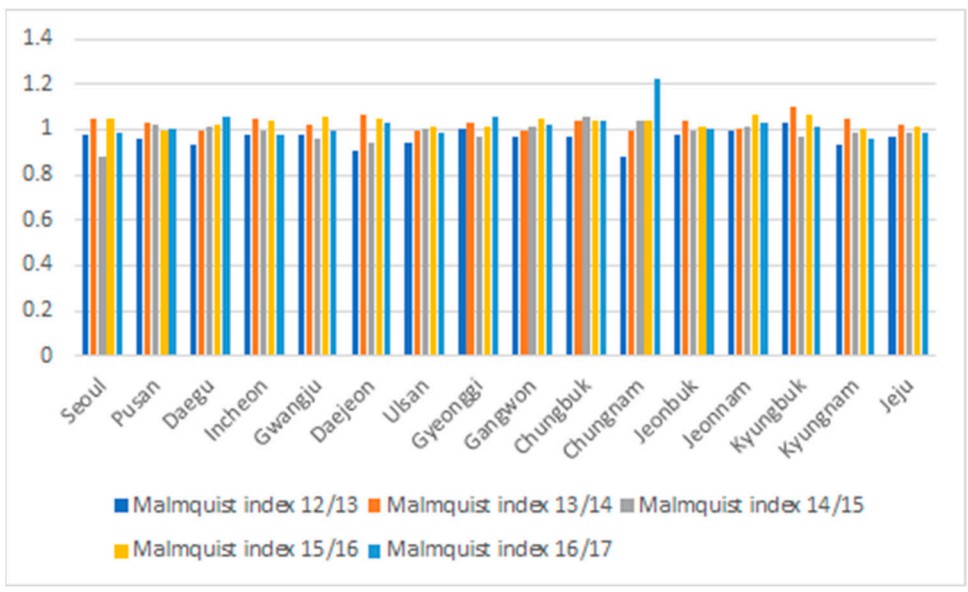

**Figure 5.** Productivity trend in local governments in 2012–2017.

*4.2. Empirical Results of the Tobit Analysis*

At the second stage of our research, we evaluated the potential factors promoting the sustainable performance of air pollution policies. As shown in Table 5, we found that economic development, represented by GRDP, is the most outstanding determinant for promoting $PM_{2.5}$ efficiency in terms of its statistical significance and its estimated size of coefficient. Regulation on the greenbelt showed a negative effect on $PM_{2.5}$ efficiency, but it is statistically not significant. Policies on renewable energy were statistically significant, but had negative effects on $PM_{2.5}$ efficiency. Population density showed a statistically acceptable and positive effect on air pollution mitigation, implying that local governments must avoid urbanization, which results in heavier air pollution. Innovation promotion policies, represented by the patent number, showed statistically acceptable and positive effects on $PM_{2.5}$ efficiency.

**Table 5.** Tobit regression result.

| Explanatory Variables | Unit | Coefficient | Std. Err. | t | $p >$ \|t\| |
|---|---|---|---|---|---|
| GRDP | Million Won | 0.0148 | 0.11 | 18.26 | 0.000 |
| Greenbelt width | $m^2$ | −1.54 | 1.26 | −1.22 | 0.227 |
| Renewable Energy generation | toe | −9.30 | 1.21 | −7.67 | 0.000 |
| Population | 1000 people/$km^2$ | 0.0171 | 4.92 | 3.46 | 0.001 |
| Patent | EA | 3.63 | 0.026 | 2.39 | 0.019 |

*4.3. Discussion*

In the first stage of our efficiency evaluation focused on the performance of local governments, Seoul and Ulsan remained efficient throughout the study period, indicating its effective environmental management, while holding significant economic volume in our research as well [4,5]. As shown in Figure 4, the $PM_{2.5}$ efficiency has increased over time, particularly after 2015, representing the turning point for the Korean government in their emphasis on clean and green policies. Companies had to adhere to these policies more actively than to the carbon emission policies. This may be because air pollution, measured by $PM_{2.5}$, is much easier for the public to monitor than invisible carbon emission, so companies are more likely to follow these policies.

However, Table 4 shows that the performance of $PM_{2.5}$ efficiency differs widely, implying that locational factors and the level of regulation in each municipality may result in differing performances.

The provinces of Chungnam and Jeonnam have more fossil-fuel power-generating companies, resulting in heavy $PM_{2.5}$, without many compensatory economic activities due to power generation policies. The result is similar for Gyeonggi, a suburb area of Seoul with comparatively low $PM_{2.5}$ efficiency, but for different reasons. The Gyeonggi province produces many environmental-unfriendly goods, while Seoul assembles environmental-friendly products, resulting in heavy transportation effects and significantly lower $PM_{2.5}$ efficiency in Gyeonggi. These kinds of environmentally harmful exports from developed to less developed areas or countries are ubiquitous [17].

Economically leading provinces, such as Seoul and Ulsan, show the best efficiency, with a value of one, while the surrounding areas, such as Gyeonggi and Kyungnam, have a much lower performance in $PM_{2.5}$ efficiency. Therefore, the contents of air pollution should be considered. Most $PM_{2.5}$ from Gyeonggi and Kyungnam consists of nitrogen oxide because of heavy cargo transportation to and from Seoul and Busan. This suggests that a more customized response is required to mitigate air pollution. Moreover, the $PM_{2.5}$ efficiency that reflects the clean air performance level of the 16 Korean local governments has a similar pattern to that of their greenhouse gas performances (green economy) [4]. For example, Seoul and Ulsan show the best performance in clean and greenhouse gas efficiency, and Jeonnam and Jeonbuk have the worst performance in both clean and green policies [4]. This result implies that achieving both green and clean economy is not paradoxical, so the Porter hypothesis could be adopted for both targets, without decoupling. However, as we can see from the performance of Jeju, green and clean performances do not always show a coupling trend, but can have different patterns due to local conditions, such as the governance or competitiveness of policies. Therefore, more customized or local economy-friendly policies are required to reflect those regional characteristics precisely and appropriately.

In Figure 5, productivity, measured by MPI, dropped rapidly in Seoul, particularly in 2015, showing the lack of governance in air pollution policies, with a relatively low performance over time. It implies that the Porter hypothesis does not always work, and the coupling between economic performance and clean air does not go together. It is interesting that Chungnam showed a strong peak in productivity growth in 2017, with its low $PM_{2.5}$ efficiency, implying another decoupling issue. This may result from the positive effect of the promotion of Sejong as an administrative capital in 2017, with many government offices moving from Seoul to Sejong, a center of Chungnam. Due to this unexpected boom in economic development in Sejong, the suburban areas of Chungnam could experience a high peak in productivity. Therefore, Figure 5 concludes that there is no sustainable performance of air pollution policies in Korean local governments because most municipalities lack governance in policies.

Now, looking to the empirical results of the second stage, the Tobit analysis shows the variable of greenbelt policy is rejected, implying a lack of governance on greenbelt policies, while the GRDP and other variables are accepted, implying a good governance of regulatory policies. For GRDP, it is positive and significantly affected by clean air, implying a decoupling effect of air pollution regulation. This explains why Seoul performs wells in its air pollution policies. Due to this overwhelming effect of economic development, regulation of the greenbelt showed a negative effect on $PM_{2.5}$ efficiency with a negative sign, but it is statistically not significant, implying that greenbelt regulation in the suburban areas of metropolitan cities does not affect the governance of environmental protection. The negative sign implies that a local government must seriously reconsider the role of greenbelt for future challenges. The reason for the rejection of the regulatory greenbelt policies on $PM_{2.5}$ efficiency may come from the fact that most local governments manage greenbelts for their administrative convenience, rather than for its potential benefits and environmental protection. The problem is not the greenbelt system itself, but the practices

that do not work out as planned. Policies on renewable energy were statistically significant but had negative effects on $PM_{2.5}$ efficiency because of the same reason as the failure of greenbelt policies. The local governments should push companies to follow transparent and predictable policy strategies. Both policies show a lack of governance on the part of local governments in terms of greenbelt and renewable energy policies. For the role of patents, innovation promotion policies, represented by the patent number, showed statistically acceptable and positive effects on $PM_{2.5}$ efficiency, implying that promotion policies on research and development always aid future strategies. Nonetheless, its estimated coefficient is small to negligible. Therefore, to boost the effect of R&D on air pollution, more customized incentives and promotion policies are required. For example, the matching fund type of initial seed money, as well as the performance-based pyramid type of R&D support, would accelerate more field- and performance-oriented governance.

## 5. Conclusions

Numerous studies have examined the environmental and energy (E&E) field related with urbanization issues, but very few studies have focused on polluted air. Nonetheless, urbanization has been a challenging political issue in developing countries. Air pollution in metropolitan areas has become so severe that local governments need to emphasize the importance of regulatory policies. Unfortunately, in most cases, there has been a lack of governance focused on mitigating air pollution. Most research emphasizes the global climate crisis. This study considered the paradoxical objective of local governments to align economic development and environmental protection with a green economy based on carbon emission mitigation and clean air based on $PM_{2.5}$ regulation. The most important contribution of this study is based on the new implication that the Porter hypothesis is achievable on clean air policies. The more urbanization develops, the more possible is local government's cleaner performance.

In the empirical results, due to the lack of governance in both greenbelt regulation and renewable energies, we showed some examples of provinces decoupling between economic development and environmental protection. Unfortunately, most Korean local governments could not harmonize the two different directions of policy. However, alignment is achievable via coupling. To do so, the most important strategic governance factor is the field-oriented customization of green and clean policies, reflecting local conditions more specifically and accurately. This is the study's second contribution. There is no panacea to achieve clean and green urbanization. To boost the green and clean sustainable economy, this study confirmed that policies should be more customized according to localization factors, and the effectiveness of policies with regard to companies depends on field- and performance-oriented incentives and other political support.

The most important finding on regulatory policies for sustainable development is to provide a transparent and predictable path for the future, because many local government policies, such as greenbelt regulation, have not been sustainable due to the unexpected change of policies, such as on urbanization or the deregulation of greenbelt. Even if the government emphasizes environmental protection on greenbelt or renewable energy, if these regulatory policies are implemented from an administrative convenience perspective, the proactive participation of companies cannot be achieved.

Therefore, for local governments in developing countries to achieve green and clean economic development, the public and private sectors should consider these issues according to the aligned perspectives, in a coupling manner. Moreover, air pollution may emerge because of diverse reasons and, thus, inclusive cooperation between the public and private sectors and between neighboring countries should be promoted.

Although this study offers very practical implications, there are some limitations. First, because the $PM_{2.5}$ emission data are not available after the year 2017, we were only able to reflect on six years, which is not enough to obtain implications of time change or policy implementation. Therefore, more recent data are required to overcome this problem. Second, as this study takes a non-parametric approach, it is recommendable to conduct

bootstrapping or to select a stochastic model to increase the statistical reliability. Last, because air pollution is caused by various factors, we may obtain more implications by adopting other pollution variable, such as NOx, Sox and O3. Urbanization drawbacks may result from diverse reasons, and some of those reasons conflict with each other. Fortunately, we found that green and clean air policies can promote economic development, but in reality, to drive economic development, the deregulation of greenbelt resulted in an unexpected air pollution mishap, resulting in a lack or weakening of governance. Therefore, more research is needed to find out how to harmonize these urbanization issues.

**Author Contributions:** Conceptualization, Y.C.; methodology, H.L. and J.D.; validation, Y.C. and H.L.; investigation, Y.C. and H.L.; data curation, H.J. and J.D.; writing—original draft preparation, Y.C.; visualization, H.J.; funding acquisition, Y.C. All authors have read and agreed to the published version of the manuscript.

**Funding:** This research was supported by Inha University (67922).

**Data Availability Statement:** Not applicable.

**Conflicts of Interest:** The authors declare no conflict of interest.

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
