# Peer review of "Urbanization Paradox of Environmental Policies in Korean Local Governments"

_land, doi:10.3390/land12020436_

Round 1

Reviewer 1 Report

This study discusses the carbon emissions and air pollution of urban development from the perspective of PM2.5 efficiency. The authors are presenting detailed statistics to achieve a quantitative assessment. While some findings are listed, the manuscript requires to address the following issues to make it more accessible to the reader. Specifically,

1) The author should give the accurate source of the data or judgment conclusions that appear in the first paragraph of the introduction.

2) The literature review is not enough. I suggest to divide it into subtitles and supplement the content.

3) Section 3 should be divided into two sections: method and result. 3.1-3.3 are methods and data. 3.4 and 3.5 are results and discussions. The calculation and analysis results should be clearly presented. Then the authors should conduct in-depth discussion based on the results. However, the relevance between the discussion in the current article and the research results is not clear.

4) It is suggested that the author supplement the limitations of the study in the discussion or conclusion.

Author Response

This study discusses the carbon emissions and air pollution of urban development from the perspective of PM2.5 efficiency. The authors are presenting detailed statistics to achieve a quantitative assessment. While some findings are listed, the manuscript requires to address the following issues to make it more accessible to the reader. Specifically,

1) The author should give the accurate source of the data or judgment conclusions that appear in the first paragraph of the introduction.

Response: We made more precise and accurate graphs (Fig1,2) and following paragraph you mentioned is written based on webpage (https://kosis.kr/). We added this address as reference 3. In addition we changed starting paragraph (introduction) as follows.

--> Until the mid-2010s, Korea has never had significant air pollution issues, so the focus of the environmental policies has always been on the abatement of greenhouse gases (GHG) [3]. As shown in Figure 1, till 2014, air pollution was low and stable, resulting in no emphasis on the clean air.

2) The literature review is not enough. I suggest to divide it into subtitles and supplement the content.

Response: We added subtitle (2.1, 2.2) for environmental efficiency and tobit regression respectively. At the same time, we added more reference and paragraph. Please see highlight in literature review section.

3) Section 3 should be divided into two sections: method and result. 3.1-3.3 are methods and data. 3.4 and 3.5 are results and discussions. The calculation and analysis results should be clearly presented. Then the authors should conduct in-depth discussion based on the results. However, the relevance between the discussion in the current article and the research results is not clear.

Response: Based on your comment, we divided section into two chapter: Ch3. Methodology and Data, Ch,4 Results and Discussion (Please see newly revised section 3, 4). Because of newly made discussion section, we added more complementary paragraph for policy implication in detail in section 4.

4) It is suggested that the author supplement the limitations of the study in the discussion or conclusion.

Response: We have added the limitations of the research at the end of conclusion section.

--> Although this study offers very practical implications, there still exists limitations. First, because PM2.5 emission data is not available after the year 2017, we only reflected short 6 years, which is not enough to obtain implications coming from time change or policy implementation. Therefore, recent data is required to overcome this problem. Second, since this study takes non-parametric approach, it is recommendable to conduct bootstrapping or to select stochastic model to increase statistical reliability. Last, because air pollution is caused by various factors, we may obtain more implications by adopting other pollution variable such as NOx, Sox and O3. Urbanization drawbacks may result from diverse reasons, and some of those are conflicting each other. Fortunately, we found the green and clean air policies can promote together with economic development, but in reality to drive economic de-velopment stronger, reckless deregulation on greenbelt resulted in the unexpected air pollution mishap, resulting in lack or weakening of governance. Therefore, more research needs to find out this harmonization on the urbanization issues.

Reviewer 2 Report

I have several concerns regarding the submission:

1) I doubt the novelty of the paper. The current submission looks like the authors' work published in Sustainability (https://www.mdpi.com/2071-1050/12/20/8359). The literature review and regression model are very similar in the two pieces of work though the subject countries under investigation are different. 

2) The authors should have paraphrased or rewritten the sentences extracted from their own previous works.  Please find the iThenticate report in the attachment for information. Otherwise, the authors may be accused for self-plagiarism.

3) For the DEA, I doubt if "energy consumption" should be taken as an input.  It seems to me that "energy consumption" should be one of the output items. If the authors really want to proxy input factors like economic activities, they should use GDP or some similar indicators at the local government levels.

4) What are the limitations of the research?  How did these limitations affect the interpretation of the findings of the research?

5) In the discussion of the research findings, the authors should compare the findings of the research with those of the previous studies. For example, is the PM2.5 efficiency in Korean provinces different from (or higher than) other jurisdictions? Are the determinants of PM2.5 efficiency in Korea different from those identified in other parts of the world by previous studies.

6) The conclusion "the most important finding on regulatory policies for sustainable development is to provide a transparent and predictable path for the future" is not justified or supported by the findings of the current research.

Furthermore, some minor edits are needed for the paper. For example, axes in Figures 1 and 2 were not properly labelled. Also, there are some grammatical errors in the paper. Besides, words like "green belt" and "greenbelt" are not used consistently through the paper.

Author Response

I have several concerns regarding the submission:

1) I doubt the novelty of the paper. The current submission looks like the authors' work published in Sustainability (https://www.mdpi.com/2071-1050/12/20/8359). The literature review and regression model are very similar in the two pieces of work though the subject countries under investigation are different. 

Response: Although our study took similar approaches with the paper (https://www.mdpi.com/2071-1050/12/20/8359), we made “Unique Differentiation” in two perspectives.

First, this study is differentiated in that it focuses not on the green air policy represented in many environmental papers, but on the air pollution issues represented by PM 2.5, resulting in the new implication that the Porter hypothesis is achievable even on clean air policies. Almost all papers in this filed take CO2, or greenhouse gas to draw implications on climate change, global warming. However, to the best of our knowledge, very few paper took PM2.5 emission as undesirable output especially, provincial level of Korea.

Second, when it comes to methodological approach, we selected five variables on the ground of previous studies to compare with other environmental papers focused on the CO2. Many papers argued that the environmental policy is feasible with economic development, and our paper share the same conclusion with clean air policies. But out unique finding is on that this coupling between clean air and economic development needs to be more customized governance. We showed it by the role of greenbelt policy on clean and green air policies.

Therefore, we believe that topic of our study is worthy of investigating by the common and unique contributions of the implications.

2) The authors should have paraphrased or rewritten the sentences extracted from their own previous works.  Please find the iThenticate report in the attachment for information. Otherwise, the authors may be accused for self-plagiarism.

Response: We got iThenticate report from editor, and did our best to revise similar parts from our previous works. If it still needs further revision for this concern, we are willing to revise it again or use English proof service again. (We attached English proof certificate)

3) For the DEA, I doubt if "energy consumption" should be taken as an input. It seems to me that "energy consumption" should be one of the output items. If the authors really want to proxy input factors like economic activities, they should use GDP or some similar indicators at the local government levels.

Response: In traditional studies using concept of DEA, labor and capital were used as inputs and GDP or sales revenue were used as outputs. Nowadays, “energy consumption” and “CO2 or PM2.5” have been widely used as additional input & output values to explore environment efficiency/productivity. Therefore, we adopted labor, capital and energy consumption as input values, and GRDP an PM2.5 emission as output values. Please see related articles as follows.

  1. Yang, Fan, Yongrok Choi, and Hyoungsuk Lee. "Life-cycle data envelopment analysis to measure efficiency and cost-effectiveness of environmental regulation in China’s transport sector." Ecological Indicators 126 (2021): 107717.
  2. Zhang, N.; Choi, Y. Environmental energy efficiency of China’s regional economies: A non-oriented slacks-based measure analysis. The Social Science Journal 50, 2013, 225-234.

4) What are the limitations of the research?  How did these limitations affect the interpretation of the findings of the research?

Response: We have added the limitations of the research as follows.

--> Although this study offers very practical implications, there still exists limitations. First, because PM2.5 emission data is not available after the year 2017, we only reflected short 6 years, which is not enough to obtain implications coming from time change or policy implementation. Therefore, recent data is required to overcome this problem. Second, since this study takes non-parametric approach, it is recommendable to conduct bootstrapping or to select stochastic model to increase statistical relia-bility. Last, because air pollution is caused by various factors, we may obtain more implications by adopting other pollution variable such as NOx, Sox and O3. Urbanization drawbacks may result from diverse reasons, and some of those are conflicting each other. Fortunately, we found the green and clean air policies can promote together with economic development, but in reality, to drive economic development stronger, reckless deregulation on greenbelt resulted in the unexpected air pollution mishap, resulting in lack or weakening of governance. Therefore, more research needs to find out this harmonization on the urbanization issues.

5) In the discussion of the research findings, the authors should compare the findings of the research with those of the previous studies. For example, is the PM2.5 efficiency in Korean provinces different from (or higher than) other jurisdictions? Are the determinants of PM2.5 efficiency in Korea different from those identified in other parts of the world by previous studies.

Response: As we explained in comment 1 & 2, this is the first study which takes PM2.5 emission in local governments level in Korea. Therefore, it was difficult to compare with previous PM2.5 efficiency of Korean local governments. Nonetheless, in order to compare and find out similar as well as unique implications with environmental papers used CO2, we used similar inputs in our model with different output of PM2.5. Moreover, regarding determinants of PM2.5 efficiency, we added more citations and added paragraph to support feasibility of dependent variables choice.

6) The conclusion "the most important finding on regulatory policies for sustainable development is to provide a transparent and predictable path for the future" is not justified or supported by the findings of the current research.

Response: We explained the background of this finding as “because many local government policies such as greenbelt regulation was not sustainable due to the unexpected change of policies on urbanization to deregulation on the greenbelt” This shall show the direct relationship between predictable policies and its governance.

Furthermore, some minor edits are needed for the paper. For example, axes in Figures 1 and 2 were not properly labelled. Also, there are some grammatical errors in the paper. Besides, words like "green belt" and "greenbelt" are not used consistently through the paper.

Response: We remade graphs with same unit (1 ton) and we have revised all errors you mentioned, and found the rest of errors across the manuscript.

Round 2

Reviewer 1 Report

All my comments at this stage have been answered. Thanks to the authors. Good luck. 

Reviewer 2 Report

I think the authors have addressed my comments satisfactorily.  I don't have any further comment on the paper.  Well done!